# Multiple Skeletal Anomalies of Sprague Dawley Rats following Prenatal Exposure to *Anastatica hierochuntica,* as Delineated by a Modified Double-Staining Method

**DOI:** 10.3390/children9050763

**Published:** 2022-05-23

**Authors:** Siti Rosmani Md Zin, Mohammed Abdullah Alshawsh, Zahurin Mohamed

**Affiliations:** 1Department of Anatomy, Faculty of Medicine, Universiti Malaya, Kuala Lumpur 50603, Malaysia; 2Department of Pharmacology, Faculty of Medicine, Universiti Malaya, Kuala Lumpur 50603, Malaysia; alshaweshmam@um.edu.my (M.A.A.); zahurin@um.edu.my (Z.M.)

**Keywords:** *Anastatica hierochuntica*, anatomy, congenital anomaly, double skeletal staining, skeletal development, bone

## Abstract

*Anastatica hierochuntica* (*A. hierochuntica*) is a plant that originates from Middle Eastern countries. This herb is commonly consumed by pregnant women to ease the process of childbirth. However, consumption of *A. hierochuntica* during the prenatal period may disrupt foetal development. In this study, we aimed to evaluate the potential effects of four different doses (0, 250, 500 and 1000 mg/kg) of *A. hierochuntica* aqueous extract (AHAE) on the skeletal development of Sprague Dawley rat foetuses. The AHAE was administered from gestational day (GD) 6 till GD20. We also aimed to produce a simplified and reproducible skeletal staining procedure for proper skeletal assessment of full-term Sprague Dawley rat foetuses. Skeletal structures were stained using a modified method that utilised Alcian Blue 8GX and Alizarin Red S dyes. The staining procedure involved fixation, skinning, evisceration, cartilage staining, bone staining and clearing. Our modified staining technique has successfully showed a clear demarcation between the bone and cartilage components, which enabled objective assessment of the skeletal ossification following administration of AHAE. Some skeletal anomalies such as sacrocaudal agenesis and maxillary defect (cleft lip) were observed in 250 and 1000 mg/kg groups, respectively. These findings indicate potential toxicity effects of AHAE on the developing foetuses.

## 1. Introduction

*Anastatica hierochuntica* L. (*A. hierochuntica*) belongs to the family Brassicaceae, and is the only member of the genus Anastatica [1]. Its name has been confirmed with http://www.theplantlist.org (last accessed on 18 July 2020). The countries of origin include Saudi Arabia, Egypt, Jordan, Iraq, the United Arab Emirates, Iran, Kuwait and North Africa [2]. In Arab countries, *A. hierochuntica* is known as Kaff Maryam [2]. In Malaysia, the plant is named as Sanggul Fatimah [1]. It is the most commonly consumed medicinal plant among pregnant women in Malaysia [3], despite the lack of scientific evidence regarding its potential toxic effects. In spite of being non-local, its supply has always been adequate to meet the local demand. It is consumed orally in the form of infusion or fine powder [1,2]. The infusion form is prepared by soaking the dried plant in water, which causes the initially shriveled branches to expand and straighten [1].

This plant is traditionally consumed due to its purported medicinal benefits and is used in the treatment of various diseases including diabetes mellitus, uterine haemorrhage, as well as to ease the process of labour [2,3,4]. In fact, its crude extract has been demonstrated to have antioxidant, antimicrobial, antimelanogenic, hypoglycaemic, hypolipidaemic and anti-inflammatory activities [5,6,7]. In spite of being widely accepted and consumed by people across the globe, very few studies have evaluated its potential side effects [8]. A study conducted by Kim and Lean (2013) reported that 63.9% out of 460 respondents (Malay pregnant women) consumes *A. hierochuntica* during gestation, particularly close to delivery because they believed that it eases the process of childbirth [3]. However, this herb has also been used during early stages of pregnancy due to its beneficial properties [3]. According to Kim and Lean (2013), majority of women in their study consumed one glass of herbal tea per day and some of them were unsure of the amount they consumed daily [3]. The presence of bioactive components and increasing trend of *A. hierochuntica* consumption among Malaysians without proper guidelines of the safe dosage, especially for the pregnant women [3] indicates a need for a thorough study investigating their potential toxicity effects on the mother as well as the developing foetus. Consumers should be alerted to the possible risks that may arise following consumption of this plant. The findings from this study may help the health care professionals to educate pregnant women about the potentially toxic effects of consuming *A. hierochuntica*.

Literature data on the effects of *A. hierochuntica* on development of the skeleton are also very scarce. In studies of the effect of external factors on prenatal development, skeletal ossification is considered as an indicator of maturity [9]. Skeletal maturity is assessed by recording the ossified, non-ossified and incompletely ossified parts of the skeleton, as well as the presence of skeletal anomalies [9]. Studies have shown that some herbal medicines have been proven to delay the ossification process, thus resulting in stunted or abnormal shapes of the bones [10,11]. The reduction in midfacial growth is one of the skeletal malformations that can occur during prenatal development. It is associated with maxillary hypoplasia, as observed in the cleft lip and palate (CLP) [12]. The prevalence of CLP in Malaysia is approximately one in 941 live birth each year [13]. The aetiology of CLP in Malaysia is multifactorial [14,15,16]. The Ministry of Health of Malaysia has conducted various health promotion programmes and initiatives to increase women’s awareness of the risk of exposure to environmental factors such as tobacco, alcohol, poor nutrition, medicinal drugs and teratogens, and lack of folic acid during pregnancy. Therefore, this study aims to evaluate the effects of *A. hierochuntica* aqueous extract (AHAE) on skeletal development of rat foetuses using a modified double skeletal staining method. One of the assessed bones is the maxilla which is associated with the development of cleft palate.

The staining method of the skeleton was originally developed in 1897 by Schultze [17]. Since then, the method has been modified by many researchers [18,19,20,21,22,23,24,25]. Later, an alternative stain, toluidine blue, was introduced for cartilage staining [26,27,28], followed by the use of Alcian Blue in 1970 by Ojeda and team [29]. Employment of Alcian Blue in staining cartilages of a mouse was initially practised in 1976 [30]. This method was then modified and employed in subsequent studies [31,32,33].

The previous methods require tedious process of frequent and multiple changes of solution throughout the staining procedure. We aim to simplify this laborious method and, at the same time, improve the consistency of the staining results. An efficient, simplified and reproducible method is important for our developmental toxicity studies because of the numerous specimens involved. Therefore, this study aims to introduce a modified skeletal double-staining technique. The null hypothesis was “aqueous extract of *A. hierochuntica* is safe and has no prenatal developmental toxicity effects on the skeletons of Sprague Dawley rats”.

## 2. Materials and Methods

### 2.1. Preparation of Anastatica hierochuntica Aqueous Extract

*Anastatica hierochuntica* was imported from Saudi Arabia. The plant voucher sample was deposited in the herbarium of University of Malaya (UM) with voucher number KLU49457, as determined by a botanist. The aqueous extract of *A. hierochuntica* was prepared by mixing the whole plant with sterile double-distilled water (ddH_2_O) (1:10). The mixture was then left in a shaking water bath at 60 °C for 4 h and filtered twice using Whatman No. 1 filter paper. The filtrate was subsequently freeze-dried to form powder. The aqueous yield extract was 4.6%.

### 2.2. Animals

The experiment was performed according to the guidelines for proper ethical use of laboratory animals (2014-03-05/ANAT/R/SRMZ), and conducted at the Association for Assessment and Accreditation of Laboratory Animal Care (AAALAC) international accredited laboratory. Male (used for mating purposes only) and female (n = 8 per group) Sprague Dawley (SD) rats aged 8 to 9 weeks with 190 to 250 g body weight (BW) were purchased from the University of Malaya (UM) Animal Experimental Unit (AEU).

### 2.3. Chemicals

The chemicals used in this study were ketamine, xylazine, 10% buffered formalin, absolute ethanol (EtOH), 95% EtOH, 70% EtOH, glycerin (Merck, Darmstadt Germany), Alizarin Red S (CAS NO. 130-22-3, R & M Marketing, Essex, UK), Alcian Blue 8GX (CAS NO. 33864-99-2, Sigma-Aldrich. St. Louis, MO, USA) and potassium hydroxide (KOH) (CAS No. 1310-58-3).

### 2.4. Experimental Design

#### 2.4.1. Mating

Prior to mating the animals, the phases of the oestrus cycle of female SD rats were determined by examining the type of cells in their vaginal smears. Vaginal fluid was obtained using plastic pipette filled with 10 µL of normal saline (NaCl 0.9%). Female rats at the proestrus phase were identified by the presence of predominance of epithelial cells in the vaginal smears under microscope. Each of these female rats was then caged on a one-to-one basis overnight with a male rat (aged 8 to 10 weeks). Vaginal smear examination was performed on the next morning to look for the presence of sperms. In the presence of sperms in the smear, the rat was considered to be pregnant at gestational day (GD) 0.

#### 2.4.2. Administration of *Anastatica hierochuntica* Aqueous Extract

The pregnant rats (n = 32) were randomly assigned into four groups, namely, one control group (the rats were administered with ddH_2_O only) and three experimental groups which were treated with *A. hierochuntica* aqueous extract (AHAE) (250, 500 and 1000 mg/kg AHAE). Administration of ddH_2_O and AHAE began from GD6 until GD20 using an oral gavage tube at 5 mL/kg. The doses were determined based on a preliminary study.

#### 2.4.3. Caesarian Section

Pregnant female rats were then anaesthetised at full-term of gestation (GD21) using ketamine (50 mg/kg) and xylazine (5 mg/kg), followed by caesarean section to collect the foetuses. The dams were then euthanised via exsanguination of the cardiac artery. Following that, the foetuses were euthanised using an overdose of ketamine and xylazine. Half of the number of foetuses from each litter were included for this skeletal study. Another half of them were subjected for brain assessment.

#### 2.4.4. Modified Double Skeletal Staining Procedure

##### Fixation of Full-Term Foetuses

The euthanised foetuses were fixed by immersion in 10% buffered formalin. After 24 h, the specimens were transferred to 70% EtOH for further fixation until the commencement of the staining process (Table 1).

##### Skinning and Evisceration

Skin removal (Figure 1) was performed by creating a vertical slit and a horizontal slit of the skin over the mid-dorsal region by using micro-dissecting spring scissors. Firstly, a vertical mid-line slit was made at the dorsal lumbar region, then the slit was extended upward to the tip of the nose and downward to the tail respectively. Then, a horizontal slit was made on the dorsal surface of the forelimb and another on the dorsal surface of the hindlimb. The slits were extended from the mid-dorsal region to the distal end of the limbs.

Subsequently, the mid-dorsal skin edges were pulled gently apart from the body by using micro-dissecting forceps towards the ventral surface of the rat. The remaining skin of the hands, feet and tail were also removed as much as possible. Since it was not practical to remove all the skin from the digits, the skin on the dorsum of each toe was snipped off using micro-dissecting spring scissors or micro-dissecting forceps up to the proximal interphalangeal joint. The interscapular fat pad was also manually removed.

Another horizontal incision was made on the anterior abdominal wall at the suprapubic level using micro-dissecting spring scissors. This incision was to create an opening for the removal of the viscera. For this purpose, a pair of micro-dissecting forceps was inserted through the opening to remove the various organs including the trachea, oesophagus and all the other organs of the thorax, abdomen and pelvis.

##### Dehydration

Once evisceration was completed, the foetuses were immersed in 95% EtOH for the initiation of the dehydration process. Dehydration was achieved using ascending concentrations of EtOH (95% EtOH followed by absolute EtOH for approximately 24 h for each step) to prepare the specimen for staining with Alcian Blue.

##### Alcian Blue Staining

Alcian 0.01% solution was prepared by dissolving 0.01 g of Alcian Blue 8GX powder in a mixture of 70 mL absolute EtOH and 30 mL of glacial acetic acid. The solution was thoroughly mixed using a magnetic stirrer. The specimens were immersed in the Alcian Blue solution for 12 to 24 h (Figure 2A).

##### Rehydration

The foetal specimens were transferred from the Alcian solution into a series of descending concentrations of EtOH for the serial rehydration process (Figure 2B). This process started with the immersion of the specimen in 95% EtOH for 1 to 2 days, followed by 70% EtOH for another 1 to 2 days, and finally, the specimens were immersed in ddH_2_O for another 1 to 2 days or until the specimens settled to the bottom of the specimen bottle. However, the presence of air bubbles in the body cavities might prevent the specimens from sinking. Therefore, the air bubbles need to be manually removed. It is also important to note that the specimens must be properly rehydrated before staining with Alizarin Red dye.

##### Alizarin Red S Staining

Alizarin solution was prepared by dissolving Alizarin Red S powder in 2% KOH to make 0.002% Alizarin solution. The well-hydrated specimens were then immersed in the staining solution for 12 to 24 h at room temperature (Figure 2C).

##### Clearing I

The specimens were subsequently subjected to a clearing process using 2% KOH after the staining procedure with Alizarin Red S. The duration required for clearing I was 12 to 24 h (Figure 2D).

##### Clearing II

Final clearing (clearing II) was performed by using a mixture of 2% KOH and 85% glycerin (1:1). It is important to note that the total duration required for clearing tends to vary according to the size of the specimen, the concentration of the clearing solution and the temperature. It is therefore crucial to note that if the specimen is left for too long in the clearing solution, the bones will disintegrate and might fall apart. In this study, the optimum duration for clearing II was established to be between 3 and 5 days when the skeletons were clearly visible.

##### Preservation

After approximately three days of immersion in a mixture of 2% KOH and 85% glycerin, the bony parts were stained velvet while the cartilage parts were stained blue. The final step was to preserve the specimens with 100% glycerin, to which thymol crystals (0.5 to 1 mg) were added to prevent fungal growth.

#### 2.4.5. Skeletal Examination

The skeleton was examined using a Dino Capture 2.0 software version 1.3.5 with Dino-Lite and Dino-Eye camera attached to a stereomicroscope (Nikon, SMZ445, Tokyo Japan). Findings from the foetuses of treated dams were compared with those of the control group. The detailed examination of the skeletal structures was performed by the principal investigator of this study who is also an anatomist, Dr. Siti Rosmani, and the findings were approved by another anatomist. The examination was carried out carefully to ensure that the structures were properly assessed and evaluated.

##### Skull and Hyoid Bone Examination

All bones of the skull including the frontal, parietal, nasal, occipital, interparietal (IP), maxilla, zygomatic and tympanic bone, as well as hyoid bone, were examined for their shape, size and degree of ossification.

##### Sternum Examination

Each segment of the sternum (sternebrae) was examined for its shape, degree of ossification and size.

##### Ribs Examination

The number of ribs was counted and all bones and cartilages were examined for abnormalities based on their shape, size and number.

##### Vertebral Column Examination

All vertebral column bones, namely, the cervical, thoracic, lumbar, sacral and caudal vertebrae were examined for the number of ossification centres (OC) and shape.

##### Forelimb and Hindlimb Examination

The number, shape and degree of ossification of the bones of the forelimb including metacarpal, radius, ulna, humerus, scapula and clavicle, and of the hindlimb including the metatarsals, tibia, fibula, femur, ilium and ischium, were examined for any abnormalities and/or malformations.

#### 2.4.6. Statistical Analysis

The data of this study were analysed and presented as percentage of occurrence. There were no specific quantitative statistical methods applied due to skeletal anomalies data variability between groups. Significance of certain skeletal anomalies observed in this study was determined based on the types of abnormalities observed in the litters and the frequency of occurrence of such anomalies based on literature reviews. This is due to the fact that some bony variations could be considered as spontaneous and not treatment-related anomalies.

## 3. Results

### 3.1. Skeletal Examination

#### 3.1.1. Modified Staining Method

The modified procedures described in this study clearly showed distinct appearance of the cartilages and bones. All the main bones of the skull, vertebrae, scapula, ribs, pelvis, upper and lower extremities and the tail bones were clearly demarcated. An overall examination of the foetal skeletal appearance showed multiple skeletal anomalies in some of the experimental animals that mostly involved the foetuses of dams treated with 250 mg/kg and 1000 mg/kg AHAE.

#### 3.1.2. Thoracic Skeletal Examination

Skeletal examination of foetuses from the control group showed normal development of sternum, ribs and vertebrae (Figure 3). Malformed sternum, ribs and vertebrae were mostly found in the foetuses of dams treated with AHAE at 250 and 1000 mg/kg (Table 2).

#### 3.1.3. Skull Examination

The skulls (Figure 4A) of the foetuses were examined for the presence of malformations such as incomplete ossification (IO) and other bone anomalies. Incomplete ossification of the interparietal (IP) bone was the main variation found in the skull examination (Figure 4B). Variations were observed in 3.3, 22.2, 12.5 and 2.56% of foetuses belonging to the control, AHAE at 250, 500 and 1000 mg/kg groups, respectively (Table 2). Another skeletal malformation was maxillary bone defect in one foetus from the 1000 mg/kg group (Figure 4C) that presented with a right cleft lip.

#### 3.1.4. Forelimbs Examination

Examination of the forelimbs revealed that 10.3% of the foetuses from the dams in the 1000 mg/kg AHAE group had shortened forelimbs (Table 2). Such an anomaly was not noted in the observed foetuses of other groups.

#### 3.1.5. Hindlimbs, Pelvic and Caudal Examination

Examination of the foetal hindlimbs showed that 10.3% of the foetuses from the dams of the 1000 mg/kg AHAE group were noted to have shortened hindlimbs and ilium abnormality (Table 2). Malformed pelvic bones were also noted in 2.78% of foetuses from the 250 mg/kg AHAE group (Table 2).

### 3.2. Summary of Skeletal Examination

Examination of the skeletal system was performed on 145 foetuses from eight dams of each group (control and AHAE-treated groups). Skeletal variations and/or malformations were mostly observed in the thoracic vertebrae. The vertebrae had a dumbbell-shaped and bipartite ossification centre (OC) (Figure 5A). These findings were seen in the control and all AHAE-treated groups, with the highest percentage of dumbbell-shaped and bipartite OC observed in the 1000 and 500 mg/kg groups, respectively (Table 2). Other variations observed include split, misshaped and unossified sternebrae (Figure 5B). A small percentage of foetuses from all AHAE-treated groups had IO of the IP bone. However, these findings were considered as spontaneous variations since they were all minor variations; furthermore, skeletal variations were also seen in 3.3% of the foetuses from the control group (Table 2).

One foetus from the 250 mg/kg group was found to have sacral and caudal agenesis (Figure 5C) associated with multiple bone anomalies including misshaped ribs, IO of the thoracic vertebrae and lumbar agenesis (Figure 5D). In addition, five foetuses from the 1000 mg/kg group were noted to have multiple bone abnormalities including misaligned and unossified sternebrae, IO of the lumbar and sacral vertebrae, and short pelvic bones (Table 2).

## 4. Discussion

The null hypothesis is rejected and the alternative hypothesis is accepted whereby aqueous extract of *A. hierochuntica* can cause toxicity effects on skeletal development of Sprague Dawley rats. Intensive research concerning the investigation of potential teratogenic effects of drugs and herbal medicine in animals has to include the evaluation of foetal skeletons [34]. This requires a simple and reproducible method to process the large number of specimens. In this study, we have successfully stained skeletal structures of our specimens using a modified staining technique (Table 1), which was developed based on the existing methods employed by various researchers [31,35,36,37,38,39].

The existing skeletal staining procedures were found to be unsuitable for our specimens. This is most probably due to the different types and sizes of specimens, or inconsistency in methods and durations of fixation. Therefore, we decided to modify and optimise the protocols to suit the full-term SD rats and the fixative methods used in our prenatal developmental toxicity study. This modified protocol is beneficial for researchers who are performing skeletal assessment studies using full-term SD rats initially fixed with 10% buffered formalin followed by 70% EtOH for temporary storage prior to skeletal staining commencement.

Most of the published protocols have been performed on specimens that were directly fixed with EtOH. The skeletal staining of the specimens was also commenced shortly after fixation, which does not involve prolonged storage. From the present study, we found that the initial fixation of the specimens in 10% buffered formalin (5 to 7 days) caused the foetus to become firmer compared to specimens that were directly immersed in 70% EtOH. Formalin-fixed specimen would be easier to manipulate, especially during the skinning process. However, it is not advisable to keep the foetus in a non-buffered formalin or neutral formalin solution, as the acidity of the solution can decalcify the bones [38]. Bone decalcification will reduce the Alizarin Red stain affinity [38,40].

In studies with a large number of specimens, the duration of tissue fixation may be prolonged. In fact, staining may often have to be conducted in batches with a limited number of specimens per batch. In such cases, the specimens can be initially fixed with formalin followed by EtOH fixation until commencement of skinning process. Careful and complete skinning is mandatory to ensure proper diffusion of the staining agent into the bones and cartilages. Some researchers used other methods for skinning such as skin maceration in 4% sodium chloride [41] or in hot water [36,37]. We found that this method is unfavourable to our study, since our aim is to obtain a firm foetus during the skinning, staining and clearing processes. Some researchers immersed the specimens in acetone to remove the fat [28,31,36]. However, we omitted this step, as our specimens were completely deskinned and thus did not have much fat.

As for the Alcian Blue dilution, we initially used 70% EtOH as recommended by previous studies [38], but this produced very poor and unsatisfying staining quality. Instead, Alcian Blue stain solution in absolute EtOH produced a better result. Nevertheless, the specimens should not be left in the solution for more than 24 h to avoid overstaining of the background tissues that could interfere with the clearing and the visibility of the skeletons. We found that Alcian Blue 8GX was a good option for cartilage staining but it denatures in acid solution over a period of time. Therefore, the solution must be freshly prepared and preferably used on the same day.

Another skeletal staining agent, Alizarin (1,2-dihydroxy-9,10-anthraquinone) [42], reacts and gives colour to the calcium deposits [40]. The colour exhibited by Alizarin Red S staining depends on the pH of the solution [40,43]. Calcium deposits are stained red by Alizarin Red S at approximately pH 9 [40]. Staining with Alizarin Red S must be carried out after the rehydration step as the stain is water-soluble. In our study, the bones were stained velvet at approximately pH 14. In our protocol, the Alizarin solution was prepared using 2% KOH, thus the duration of immersion must not exceed 24 h to prevent joint detachment.

Clearing is one of the most critical steps in the skeletal staining process. Inadequate clearing will result in poor visibility of the stained skeleton, whereas excessive clearing leads to joint detachment and increased specimen fragility. Specimens with low rigidity could not absorb enough Alizarin Red stain [38]. Some of the previous studies used an enzyme (e.g., trypsin) for tissue digestion, especially for adult specimens. However, it has been shown that KOH is the best clearing agent for embryos and early postnatal samples, as trypsin is considered to be too strong for the foetuses [38]. Some studies used 0.5 or 1% KOH for tissue clearing, but in this modified protocol, we used 2% KOH. We found that clearing with 2% KOH for approximately 6 to 7 h results in satisfactory clearing of tissues. Clearing should be done at room temperature since higher temperature may destroy the joint components [38]. However, clearing at 4 °C can be implemented if it is required to slow down the process in the case of extremely small-sized foetuses or due to logistical issues that require a delay in the staining process.

Some studies used both staining agents (Alcian Blue and Alizarin) as a mixture [31,33,38], thus resulting in high colour absorption by the specimens, which may lead to a problem in achieving adequate and proper clearing or decolourisation to remove the excess stain [38]. To achieve a simplified yet satisfying decolourisation, we omitted several steps carried out by some researchers, such as clearing in an ascending series of glycerin [31,35,36,37,38,39]. The omission of the steps saves time and indirectly reduced the financial cost of the overall staining procedure.

The modification applied in the present study has produced a clearly visible skeleton for proper assessment. Following examination, one of the foetuses belonging to the 1000 mg/kg AHAE group was found to have maxillary bone defect with a right cleft lip. Development of the secondary palate in rats consists of the same essential morphological steps as those observed in humans [44,45,46,47,48,49,50,51,52]. In human, palatal shelves commence development as processes that grow downwards from the maxillary processes. In rats, the palatal shelves appear by GD13 and subsequently grow vertically down the lateral aspects of the tongue. Around GD16 (in rats), the palatal shelves elevate to a horizontal position above the tongue and establish contact by GD16.5 [53]. The presence of the maxillary defect indicates developmental arrest of the maxilla and diminished growth of the maxillary process. This could disrupt palatogenesis, thus resulting in cleft palate.

Cleft palate is one of the birth defects that can occur due to genetic or environmental factors. Rat is the main animal model used in evaluating developmental toxicity effects (e.g., cleft palate) of specific substance because toxic responses in rats can generally be extrapolated to humans [53]. Closure of the rat palate usually occurs on GD16 to GD17 [54]. Therefore, the cleft lip and palate reported in the present study is most likely to be associated with prenatal exposure to AHAE, which takes place within the susceptible period of development. Cleft lip is the most common congenital malformation of the orofacial area [55] and has been reported to occur as a result of maternal hypoxia [56], which can develop following ion channel blockage, secondary to prenatal exposure to drugs [57]. Although there was only one case of cleft palate found in this study, this finding can be considered as an adverse effect in view of the presence of other statistically significant findings as reported in our previous publication [58]. The reported findings clearly suggest that oral administration of 1000 mg/kg AHAE during pregnancy (GD6 to GD20) can cause significant toxic effects.

Conversely, all bony variations found in the control and 500 mg/kg groups were considered spontaneous, as the percentage of occurrence was low and there were no marked treatment-related skeletal anomalies as observed in the other groups. In fact, the supernumerary ribs (classified as rudimentary ribs because their lengths are <1/2 of the ossified preceding ribs) observed in the control group were already demonstrated by a previous researcher to be temporary or a non-harmful variation should they persist in postnatal life [59].

It is interesting to note that a foetus from the lowest dose group (250 mg/kg AHAE) was found to have a short stature, caudal vertebral agenesis and other associated multiple bone anomalies. These findings indicated the presence of a low observed-adverse effect level at the 250 mg/kg dose. The anomalies observed in the foetal skeletons of the lowest and highest dose groups indicates a non-monotonous dose–-response curve (NMDRC). Such dose–response curve was hypothesised to occur due to a possible disruption in hormonal regulation [60,61,62].

Additionally, flavonoids such as quercetin, naringenin and eriodictyol have been isolated from *A. hierochuntica*, as reported by previous studies [62,63]. These flavonoids are known to act as inhibitors of aromatase activity [64,65,66,67]. Inhibition of aromatase enzyme interferes with the conversion of androgen into oestrogen, and thus could increase the level of androgen and decrease the level of oestrogen. Oestrogen plays an important role in the growth and maturation of bone. During bone growth, oestrogen is needed for proper closure of epiphyseal growth plates both in females and in males. In young skeleton, oestrogen deficiency leads to increased osteoclast formation and enhanced bone resorption [68]. A study by Migliaccio and team (1996) demonstrated that alterations in the maternal oestrogenic levels during pregnancy can influence early phases of foetal bone tissue development and subsequently result in permanent changes in the skeleton [69]. However, some studies demonstrated protective effects of quercetin on skeletal development and teratogenicity [70,71]. This contrast finding is likely due to the dose-dependent effect of quercetin or other constituents of *A. hierochuntica*.

The anomalies observed in the foetal skeletons of the highest and lowest dose groups are consistent with the other prenatal toxicity effects following *A. hierochuntica* exposure described by my team in a previous publication which indicates NMDRC [58]. The non-monotonous dose–response curve of AHAE shows the importance to determine the accurate safe dose in human prior to consumption of this natural product. Since this study has shown presence of toxicity effects at certain doses, we assume that different active and secondary metabolites are present in this extract; and perhaps some of these compounds may function in a synergistic manner. Further studies using the isolated compounds are recommended to identify the phytoconstituents responsible for the toxicity effects of *A. hierochuntica*. In addition, the usage of only one type of animal model (SD rats) has limited the generalizability of the findings to other tissues or biological models. Therefore, future studies using non-rodent animal models or other rat species would be beneficial for further clarification in this matter.

## 5. Conclusions

The findings obtained from the present study suggested that exposure to 250 and 1000 mg/kg AHAE during the prenatal period is potentially toxic to the developing foetus. Therefore, it is not advisable to consume *A. hierochuntica* during pregnancy. Apart from that, the modifications that were made to the existing skeletal staining methods have resulted in a reproducible and simplified double skeletal staining procedure, which is suitable for a skeletal developmental study. The clear appearance of the bones and cartilages has made it possible for a more distinct and reliable evaluation of ossification of the animals under study.

## Figures and Tables

**Figure 1 children-09-00763-f001:**
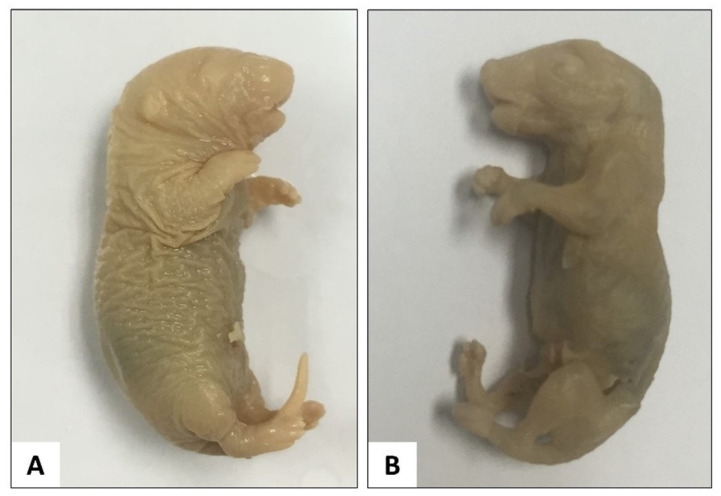
Sprague Dawley rat before skinning (**A**) and after skinning (**B**).

**Figure 2 children-09-00763-f002:**
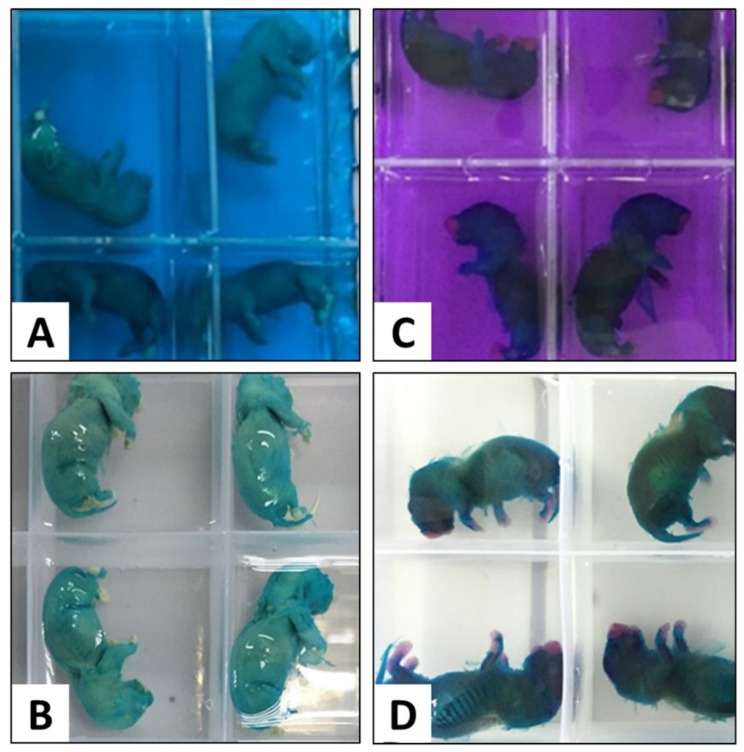
Staining of cartilage using Alcian Blue (**A**); Rehydration of foetal specimens in a descending series of concentration of ethanol (**B**); Staining of bone using Alizarin Red S solution (**C**); Clearing with KOH (**D**).

**Figure 3 children-09-00763-f003:**
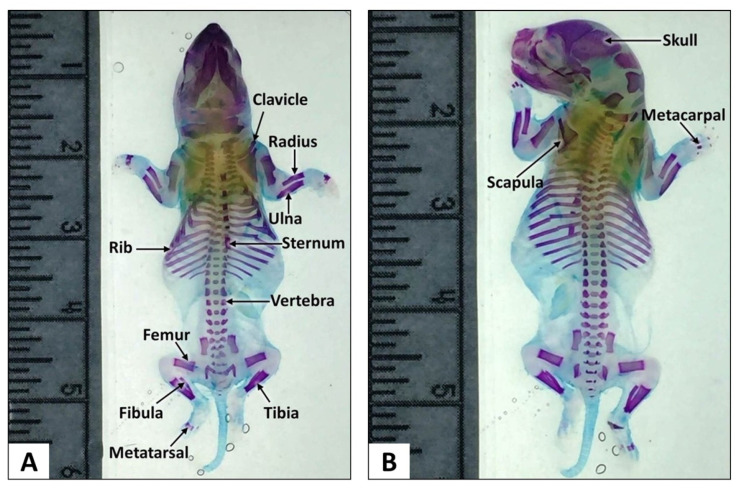
Normal skeleton of a full-term SD rat foetus stained using a modified double-staining technique for the skeletal system. Ventral (**A**) and Dorsal (**B**) views. (Scale is in cm).

**Figure 4 children-09-00763-f004:**
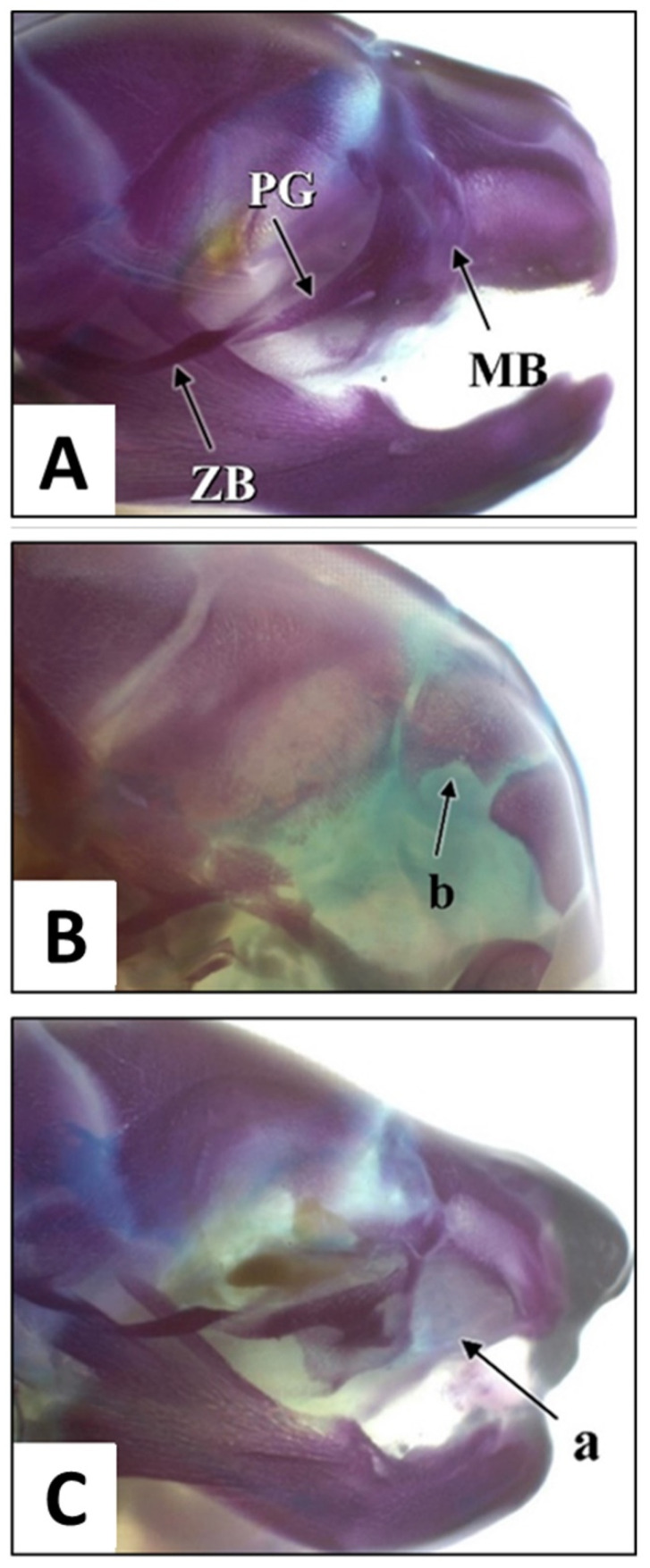
Photographs of foetal heads showing the zygomatic (ZB), processus jugalis of maxilla (PG) and maxillary bone (MB) of foetal rat. Normal ZB, PG and MB (**A**); incomplete ossification of interparietal (IP) bone indicated by ‘b’ (**B**); and maxillary bone defect (**C**), indicating presence of cleft palate (a) in a foetus from AHAE 1000 mg/kg group.

**Figure 5 children-09-00763-f005:**
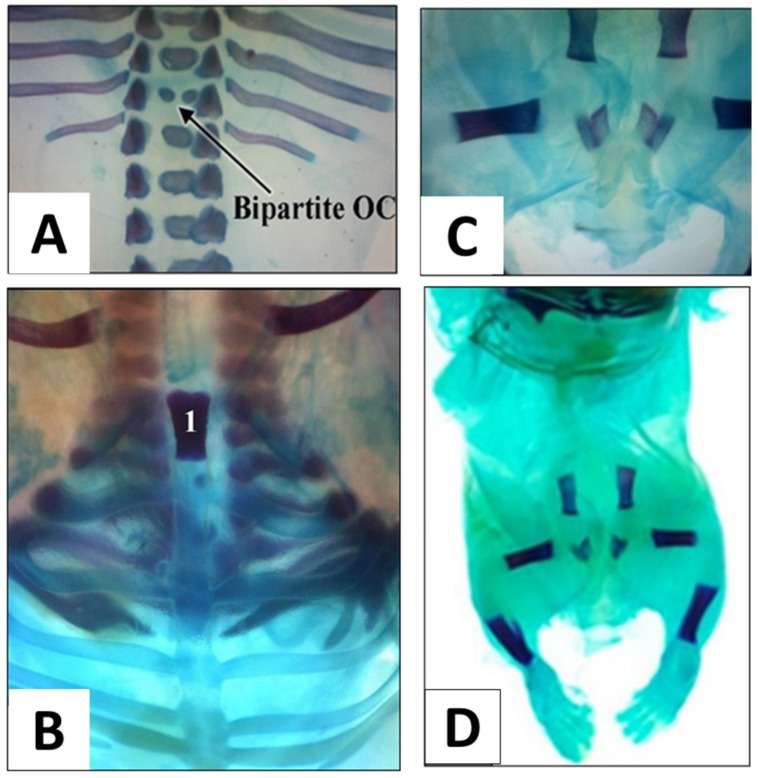
Photographs represent some of the observed skeletal abnormalities: Bipartite ossification centre (OC) of thoracic vertebra (found mostly in 500 mg/kg group) (**A**); Unossified sternebrae in a foetus of 250 mg/kg AHAE-treated group (**B**); sacrocaudal agenesis (**C**) and lumbar vertebrae agenesis (**D**) (250 mg/kg AHAE group).

**Table 1 children-09-00763-t001:** Summary of the procedures of the modified double-staining technique for the skeletal system.

No.	Step	Chemical	Duration
1	Fixation	10% buffered formalin	24 h
		70% ethanol	Until staining is commenced
2	Skinning	None	5 to 15 min
3	Dehydration	95% ethanol	~24 h
		Absolute (100%) ethanol	~24 h
4	Alcian Blue staining	Alcian Blue 8GX (0.01%)	12 to 24 h
5	Rehydration	95% ethanol	1 to 2 days
		70% ethanol	1 to 2 days
		Distilled water	1 to 2 days
6	Alizarin Red S	Alizarin Red S (0.002%)	12 to 24 h
7	Clearing I	2% KOH	12 to 24 h
8	Clearing II	2% KOH and 85% glycerin	3 to 5 days
9	Preservation	Glycerin and thymol	Storage

**Table 2 children-09-00763-t002:** Occurrence of skeletal abnormalities in the offspring of SD rats treated with different doses of AHAE.

Parameters	Group				
		Control	250 mg/kg AHAE	500 mg/kg AHAE	1000 mg/kg AHAE
Foetuses (litters) examined (N)		30 (8)	36 (8)	40 (8)	39 (8)
**Percentage of foetuses showing anomalies and (% pregnant female) in:**					
**Skull**					
Os parietale	(IO)	0 (0)	0 (0)	0 (0)	0 (0)
Os frontale	(IO)	0 (0)	0 (0)	0 (0)	0 (0)
Os occipital	(IO)	0 (0)	0 (0)	0 (0)	0 (0)
Os interparietale	(IO)	3.3 (12.5)	22.2 (50)	12.5 (50)	2.56 (12.5)
Os supraoccipitale	(misshap.)	0 (0)	0 (0)	0 (0)	0 (0)
Proc. Jugalis maxilla	(IO)	0 (0)	0 (0)	0 (0)	0 (0)
Os zygomatic	(IO)	0 (0)	0 (0)	0 (0)	0 (0)
Maxillary bone defect		0 (0)	0 (0)	0 (0)	2.56 (12.5)
**Hyoid bone**	(absent/IO)	0 (0)	0 (0)	0 (0)	0 (0)
**Sternum**					
All sternebrae	(split and misaligned)	0 (0)	0 (0)	0 (0)	2.56 (12.5)
	(unossified)	0 (0)	0 (0)	0 (0)	7.7 (12.5)
Sternebrae 1	(split manubrium)	0 (0)	2.78 (12.5)	0 (0)	0 (0)
	(IO)	0 (0)	2.78 (12.5)	2.5 (12.5)	0 (0)
	(absent)	3.3 (12.5)	0 (0)	0 (0)	0 (0)
Sternebrae 2	(misshap.)	0 (0)	0 (0)	0 (0)	2.56 (12.5)
	(IO)	0 (0)	2.78 (12.5)	2.5 (12.5)	0 (0)
	(unossified)	0 (0)	2.78 (12.5)	0 (0)	2.56 (12.5)
Sternebrae 3	(misshap.)	0 (0)	2.78 (12.5)	2.5 (12.5)	10.3 (37.5)
	(IO)	0 (0)	0 (0)	0 (0)	0 (0)
	(unossified)	0 (0)	2.78 (12.5)	0 (0)	2.56 (12.5)
Sternebrae 4	(misshap.)	3.3 (12.5)	2.78 (12.5)	2.5 (12.5)	10.3 (37.5)
	(IO)	0 (0)	0 (0)	0 (0)	0 (0)
	(unossified)	0 (0)	2.78 (12.5)	0 (0)	2.56 (12.5)
Sternebrae 5	(misshap.)	3.3 (12.5)	0 (0)	2.5 (12.5)	2.56 (12.5)
	(IO)	0 (0)	0 (0)	0 (0)	0 (0)
	(unossified)	0 (0)	2.78 (12.5)	0 (0)	5.1 (25)
	(absent)	0 (0)	8.3 (12.5)	0 (0)	0 (0)
Sternebrae 6	(split)	0 (0)	0 (0)	0 (0)	2.56 (12.5)
	(unossified)	0 (0)	2.78 (12.5)	0 (0)	0 (0)
	(absent)	0 (0)	2.78 (12.5)	0 (0)	0 (0)
Xiphisternum	(any abnormalities)	0 (0)	0 (0)	0 (0)	0 (0)
Foetuses (litters) examined (N)		30 (8)	36 (8)	40 (8)	39 (8)
**Ribs and vertebrae**					
All ribs	(misshap.)	0 (0)	2.78 (12.5)	0 (0)	0 (0)
	(Fused)	0 (0)	0 (0)	0 (0)	0 (0)
	(Wavy: 1 pair)	3.3 (12.5)	0 (0)	0 (0)	0 (0)
13th rib	(short)	0 (0)	0 (0)	0 (0)	0 (0)
14th rib/ rudimentary	(both sides)	13.3 (37.5)	5.6 (25)	10 (25)	2.56 (12.5)
	(one side)	10 (25)	8.3 (37.5)	2.5 (12.5)	0 (0)
3rd lumbar rib	(both sides)	0 (0)	2.78 (12.5)	0 (0)	0 (0)
4th lumbar rib	(both sides)	6.7 (25)	5.6 (25)	0 (0)	0 (0)
	(one side)	3.3 (12.5)	0 (0)	0 (0)	0 (0)
Atlas	(misshap.)	0 (0)	0 (0)	0 (0)	0 (0)
	(IO)	0 (0)	0 (0)	0 (0)	0 (0)
Thoracic vert. OC	(dumbbell)	16.7 (37.5)	36.1 (75)	25 (75)	41 (75)
	(bipartite)	10 (25)	13.9 (50)	17.5 (62.5)	15.4 (37.5)
	(IO)	0 (0)	2.78 (12.5)	0 (0)	0 (0)
Lumbar vert. OC	(dumbbell)	3.3 (12.5)	0 (0)	2.5 (12.5)	0 (0)
	(bipartite)	0 (0)	0 (0)	5 (25)	0 (0)
	(IO)	0 (0)	0 (0)	0 (0)	5.1 (12.5)
	(absent)	0 (0)	2.78 (12.5)	0 (0)	0 (0)
Sacral	(IO)	0 (0)	0 (0)	0 (0)	10.3 (12.5)
	(absent)	0 (0)	2.78 (12.5)	0 (0)	0 (0)
Caudal	(absent)	0 (0)	2.78 (12.5)	0 (0)	0 (0)
**Limbs and pelvic bones**					
All forelimbs	(short)	0 (0)	0 (0)	0 (0)	10.3 (12.5)
Metacarpals	(IO)	0 (0)	0 (0)	0 (0)	0 (0)
Os humerus	(IO)	0 (0)	0 (0)	0 (0)	0 (0)
Ilium	(shortened)	0 (0)	0 (0)	0 (0)	10.3 (12.5)
Pubis and ischium	(shortened)	0 (0)	0 (0)	0 (0)	5.1 (12.5)
	(unossified)	0 (0)	0 (0)	0 (0)	5.1 (12.5)
	(misshap.)	0 (0)	2.78 (12.5)	0 (0)	0 (0)
All hindlimbs	(short)	0 (0)	0 (0)	0 (0)	10.3 (12.5)
Os femur	(misshap.)	0 (0)	0 (0)	0 (0)	0 (0)
	(IO)	0 (0)	0 (0)	0 (0)	0 (0)
Tibia and fibula	(misshap.)	0 (0)	0 (0)	0 (0)	0 (0)
Metatarsals	(IO)	0 (0)	0 (0)	0 (0)	0 (0)

Values represent the percentage of foetuses (% pregnant females). Vert: Vertebrae; Os: bone; IO: incomplete ossification; Misshap: misshaped.

## Data Availability

Not applicable.

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
