# Peer review of "Multiple Skeletal Anomalies of Sprague Dawley Rats following Prenatal Exposure to Anastatica hierochuntica, as Delineated by a Modified Double-Staining Method"

_children, 2022, doi:10.3390/children9050763_

Round 1

Reviewer 1 Report

Introduction:

The topic of the manuscript is interesting, although more information should be provided, since the bibliography in the introduction is scarce. The theme needs to be improved on:
- Skeletal tissues of rodent fetuses.
- Staining technique
- Place in the world where this medicinal plant is administered/consumed.

Methodology:

Explain in more detail what the skeletal examination consists of, that is, by whom it was examined and how. Important point to understand the procedure.

Discussion

Did the research have any limitations? If so, it would be interesting to mention them.

Author Response

Response to Reviewer 1 Comments

Point 1: Introduction

The topic of the manuscript is interesting, although more information should be provided, since the bibliography in the introduction is scarce. The theme needs to be improved on:

- Skeletal tissues of rodent fetuses.

- Staining technique

- Place in the world where this medicinal plant is administered/consumed.

Response 1: Thank you for pointing this out. We agree with this comment. Therefore, we have added the following paragraphs in the manuscript.

Regarding “Places in the world where this medicinal plant is administered/consumed” a new paragraph was added (Page no. 1, Paragraph: 1  Line: 30-39)

The countries of origin include Saudi Arabia, Egypt, Jordan, Iraq, the United Arab Emirates, Iran, Kuwait and North Africa [2]. In Arab countries, A. Hierochuntica is known as Kaff Maryam [2]. In Malaysia, the plant is named as Sanggul Fatimah [1]. It is the most commonly consumed medicinal plant among pregnant women in Malaysia [3] despite the lack of scientific evidence regarding its potential toxic effects. In spite of being non-local, its supply has always been adequate to meet the local demand. It is consumed orally in the form of infusion or fine powder [1, 2]. The infusion form is prepared by soaking the dried plant in water, which causes the initially shriveled branches to expand and straighten [1].

The following sentence was deleted:

….originates from Middle Eastern countries [2] and…..

Regarding “Skeletal tissues of rodent fetuses”, we have made a short addition into the paragraph due to limited literature data (Page no. 2, Paragraph: 2, Line: 48-50)

However, literature data on the normal development of skeleton are very limited and only slight differences were noted in the prenatal ossification between different rat strains.

Regarding “staining technique”, we have made minor addition to the paragraph in the introduction section because the details have been explained in the discussion section (Page no. 2, Paragraph: 4, Line: 69-71)

In this study, we have successfully stained skeletal structures of rat specimens using our modified Alizarin and Alcian Blue staining techniques based on the existing methods employed by various researchers.

Point 2: Methodology

Explain in more detail what the skeletal examination consists of, that is, by whom it was examined and how. Important point to understand the procedure.

Response 2: Thank you for pointing this out. We agree with this comment. Therefore, we have added the following sentences (Page no. 6-7, Line: 219-246)

The detail examination of the skeletal structures was performed by the principal investigator of this study who is also an anatomist, Dr. Siti Rosmani and the findings were approved by another anatomist. The examination was carried out carefully to ensure the structures were properly assessed and evaluated.

2.4.5.1. Skull and hyoid bone examination

All bones of the skull including the frontal, parietal, nasal, occipital, interparietal (IP), maxilla, zygomatic and tympanic bone; as well as hyoid bone were examined for their shape, size and degree of ossification.

2.4.5.2. Sternum examination

Each segment of the sternum (sternebrae) was examined for its shape, degree of ossification and size.

2.4.5.3. Ribs examination

The number of ribs was counted and all bones and cartilages were examined for abnormalities based on their shape, size and number.

2.4.5.4. Vertebral column examination

All vertebral column bones namely the cervical, thoracic, lumbar, sacral and caudal vertebrae were examined for the number of ossification centres (OC) and shape.

2.4.5.5. Forelimb and hindlimb examination

The number, shape and degree of ossification of the bones of the forelimb including metacarpal, radius, ulna, humerus, scapula and clavicle, and of the hind limb including the metatarsals, tibia, fibula, femur, ilium and ischium were examined for any abnormalities and/or malformations.

Point 3: Discussion

Did the research have any limitations? If so, it would be interesting to mention them.

Response 3: Thank you for pointing this out. We agree with this comment. Therefore, we have added the following sentence in the discussion section (Page no. 15-16, Line: 473-476 )

In addition, the usage of only one type of animal model (SD rats)  has limited the generalizability of the findings to other tissues or biological models. Therefore, future studies using non-rodent animal models or other rat species would be beneficial for further clarification in this matter.

Reviewer 2 Report

The study entitled Multiple skeletal anomalies of sprague dawley rats following prenatal exposure to Anastatica hierochuntica; as delineated by a modified double staining method conducted by Siti Rosmani Md Zin and coworkers analyzed the effect of different doses of water extract of Anastatica hierochuntica growing in the Middle East on the skeletal development of sprague dawley fetuses. Skeletal abnormalities were evaluated by a modified staining method.

This study may be of interest to readers because very interesting findings are reported regarding the effect of different doses of the plant extract Anastatica hierochuntica, used during pregnancy, on the development of individual skeletal anomalies. The frequency of their distribution may have certain regularities, which is of great importance for the evidence of the obtained results of the study.

Therefore, it is recommended to include a subsection "Statistical processing of data" in the Materials and methods section, specifying methods for comparing populations of qualitative features.

In Table 2, indicate the reliability of differences in the distribution of frequencies of features, if any.

Discuss the results in the Discussion section.

Author Response

Response to Reviewer 2 Comments

Point 1: The study entitled Multiple skeletal anomalies of sprague dawley rats following prenatal exposure to Anastatica hierochuntica; as delineated by a modified double staining method conducted by Siti Rosmani Md Zin and coworkers analyzed the effect of different doses of water extract of Anastatica hierochuntica growing in the Middle East on the skeletal development of sprague dawley fetuses. Skeletal abnormalities were evaluated by a modified staining method.

This study may be of interest to readers because very interesting findings are reported regarding the effect of different doses of the plant extract Anastatica hierochuntica, used during pregnancy, on the development of individual skeletal anomalies. The frequency of their distribution may have certain regularities, which is of great importance for the evidence of the obtained results of the study.

Therefore, it is recommended to include a subsection "Statistical processing of data" in the Materials and methods section, specifying methods for comparing populations of qualitative features.

Response 1: Thank you for pointing this out. We have added the following subsection in response to the reviewers comments (Page no. 7, Line: 248-255)

2.4.6. Statistical analysis

The data of this study were analysed and presented as percentage of occurrence. There were no specific quantitative statistical methods applied due to skeletal anomalies data variability between groups. Significance of certain skeletal anomalies observed in this study was determined based on the types of abnormalities observed in the litters and the frequency of occurrence of such anomalies based on literature reviews. This is due to the fact that some bony variations could be considered as spontaneous and not treatment-related anomalies.

Point 2: In Table 2, indicate the reliability of differences in the distribution of frequencies of features, if any.

Response 2: Thank you for pointing this out. However, there is no reliability of differences in the distribution of frequencies of features, as explained in subsection 2.4.6.

Point 3: Discuss the results in the Discussion section.

Response 3: Thank you for pointing this out. The results have been discussed in the discussion section.
